# Parking Allocation Index Analysis of Office Building Based on the TOD Measurement Method

**Xiang Tang** [1], **Jianxiao Ma** [1,*], **Peng He** [2] and **Chubo Xu** [1]

1 College of Automobile and Traffic Engineering, Nanjing Forestry University, Nanjing 210000, China; xcbx@njfu.edu.cn (X.T.); stc@emails.bjut.edu.cn (C.X.)
2 Nanjing Institute of City & Transport Planning Co., Ltd., Nanjing 210000, China; hp9012@163.com
* Correspondence: majx@njfu.edu.cn

**Abstract:** Under the overall strategic guidance of emission peaks and carbon neutrality, an increasing number of cities are focusing on sustainable transportation development as an important measure for sustainable transportation development. Transit-oriented development (TOD) can guide residents to green trip options and reduce the dependence on private cars. Many cities have qualitatively reduced the parking allocation index of office buildings around rail stations, and quantitative research on the influence area and degree of TOD is lacking. This paper selects office buildings in the rail transit station influence area as the research object, puts forward the TOD measurement method of rail transit stations based on the improved "Node-Place" model, and clusters the stations under different measurement indices by the K-means algorithm. For different types of stations, the multinomial logit (MNL) model is used to build different types of trip mode split models to put forward the reduction calculation method of the parking allocation index of office buildings in the rail transit station influence area. Finally, this paper applies the revision of Nanjing's allocation index in 2019, and the TOD measurement is identified through the "Node-Place-Connection" model. The optimized calculation method of the parking allocation index for office buildings is proposed. The results indicate that the method can reduce parking allocations to encourage the use of green transportation and guide the construction of urban sustainable transportation systems.

**Keywords:** sustainable transportation; TOD measurement; rail transit stations; parking allocation; office building

## 1. Introduction

According to IEA (International Energy Agency) statistics, in China's carbon emission composition in 2018, the transportation industry produced 10% and was the third largest emission source. Under the overall strategic guidance of emission peaks and carbon neutrality, the concept of urban transit-oriented development (TOD) began to be popularized. The buildings in the TOD area advocate public transport orientation, reducing car travel and parking demand by improving rail transit accessibility, and land-use mix to control parking supply. Some solutions have been provided to support the efficient functioning of sustainable transportation development. Macioszek [1] found that the construction of P&R parking can solve road traffic problems and promote the sustainable development of urban transportation. Kitthamkesorn [2] developed a mathematical programming formulation for determining P&R facility locations; the results indicate a significant impact of route-specific perception variance on the optimal P&R facility locations. Also, some case investigations and studies in China and other countries have shown that TOD has an obvious reduction effect on building parking demand [3]. For example, Vancouver has reduced the parking allocation index of residential buildings around bus hubs by 14–28% [4], and the office building parking demand rate in the TOD development zone in California has decreased by 12–60%, compared with other areas [5]. McDonnell et al. [6] found that the parking

demand for buildings built near rail transit stations will be reduced. Tian et al. [7] combined research in 10 typical TOD areas and found that the peak parking demand for commercial buildings accounted for only 27% of the peak demand of ITEs. At present, in the parking standard documents issued by Chinese cities, such as Shanghai, Guangzhou, and Suzhou, the parking allocation index of office buildings around rail transit stations has been reduced accordingly. For instance, Shanghai proposes that in parking zone A, the parking allocation index of public buildings within a 300-m service range of rail transit stations can be appropriately reduced by less than 20%. Guangzhou advised that the parking allocation index of office buildings within 500 m of the urban rail transit station center can be reduced by 15%, and Suzhou announced that in parking zone A, the parking allocation index of office buildings located near the station entrance can be reduced, and those within 300 m of the station can be reduced by 70%.

From the current implementation in different cities in China, relatively low parking indices for office buildings are usually adopted to reasonably guide traffic travel. The definition of the scope and impact degree of TOD buildings are mainly qualitative empirical values, and the quantitative research and practical applications are relatively few.

The remainder of this paper is organized as follows: Section 2 introduces the TOD measurement model, Section 3 describes the reduction model of the parking allocation index, which were the methods used in this paper. In Section 4, we use the rail transit stations in Nanjing as an example, and the discussions of the example analysis are presented in Section 5. Finally, the conclusions drawn are presented in Section 6.

## 2. TOD Measurement Model

### 2.1. TOD Measurement

TOD is based on public transit. Its main focus is to make the rail transit hub as a core and carry out a composite layout for facilities and public space that have various purposes in the area around the station, with appropriate walking distance being the radius defining the land-use pattern with spatial compactness, high-density development, and mixed function [8]. Practice has proven that TOD is a planning method that can effectively promote urban sustainable development. It effectively improves intensive land use and the convenience of public transit, which can cultivate passenger demand for low-carbon travel. In addition, it also effectively reduces the dependence of private cars, traffic congestion, construction demand for parking spaces [9].

At present, there are more qualitative studies on the impact of public transit in TOD theoretical research and fewer quantitative studies on the scope and degree of TOD impact. Therefore, this paper proposes the concept of TOD measurement and attempts to measure the TOD development level in the area affected by rail transit stations to evaluate the TOD comprehensive development level of rail transit stations based on the measurement of different dimensions.

### 2.2. Method: Improved "Node-Place" Model

Most of the existing TOD measurement models are based on the classical "Node-Place" model or its improved model [10]. The "Node-Place" model was first proposed by Bertolini and applied to the study of rail transit station areas in Amsterdam [11]. The "Node-Place" model is based on quantitative evaluation indices to measure the degree of coordination between transportation functions and land-use levels in rail transit stations. In the model, the Node refers to the rail transit station, whose dimension is used to measure the service level of rail transit stations, and the higher the service level is, the better the regional development. Place refers to the area affected by the rail transit station. The diversity of land-use distribution in the affected area is measured by the dimension of Place, and a place with better diversity can attract higher density activity demand.

The "Node-Place" model holds that the rail transit station is not only a separate node but also a cohesive coordinative whole with the places around the station, which represents the relationship between land use and traffic development. However, this mutual feedback

relationship lacks a connecting medium. When residents arrive at the surrounding places from rail transit stations to obtain services, they need a connecting medium to characterize the accessibility of this service acquisition process, which is called "Connection" in this paper. This means that on the basis of the classical "Node-Place" model, the "Connection" dimension is added to associate the relatively isolated node transportability with the place functionality in this paper to better represent the relationship of mutual influence and restriction between Node and Place. Therefore, this paper measures TOD by establishing a "Node-Place-Connection" model.

The existing research mainly constructs the evaluation indices of the "Node-Place" model from the aspects of place service, connectivity, and degree of land-use mix. Most of the evaluation indices of the "Place" dimension address the level of land development and utilization but ignore the activities produced by streams of people in the affected area in rail transit stations. To further optimize the model, this paper introduces the "place vitality" index. Heat map data are used to characterize the performance of population flow in the spatial dimension from the affected area and measure regional vitality. In terms of the "Connection" dimension, this paper uses the application of the Walk Score [12] as a reference, and the walking index score is introduced to measure walking friendliness in a place. The Index interpretation and calculation method were shown in Table 1.

**Table 1.** Evaluation Index System of the "Node-Place-Connection" Model.

| Dimension | Measure Index | Index Interpretation/Calculation Method |
|---|---|---|
| Node | Entrances and exits | Number of entrances and exits of rail transit stations |
| | Service directions | Number of station transfer lines |
| | Departure frequency | Number of trains arriving at the station per unit time of a line |
| | Number of reachable sites | Number of stations reachable by all track lines within 20 min |
| | Scale of bicycle parking facilities | Scale of bicycle parking facilities connected to the station |
| Place | Place vitality | Heat map data about population spatial distribution |
| | Residential land area | Residential land area in the site affected area |
| | Land area for commercial service facilities | Land area for commercial, entertainment, catering and other facilities in the site affected area |
| | Land area for public management and service facilities | Land area of administrative, cultural, educational, and other facilities in the area affected by the station |
| | Industrial land area | Area of industrial land in the affected area of the site |
| | Land use information entropy | Richness and complexity of land-use types |
| Connection | Number of bus lines | Number of lines owned by all bus stops in the affected area of the site |
| | Scale of public bicycles and shared bicycles | Number of public bicycles and shared bicycles in the affected area of the site |
| | Average distance to various land uses | The average distance from the station to the residential, commercial, cultural, and other places in the affected area |
| | Walking index | The pedestrian potential of the site is evaluated by factors such as block scale and intersection density |

### 2.3. Entropy Weight—Fuzzy Comprehensive Evaluation Method

TOD measurement is a complex multifactor comprehensive evaluation process. Because the evaluation angle of subjective and objective weighting methods, such as the Delphi method, expert scoring method, entropy weight method, and coefficient of variation method, is relatively one-sided, this paper uses the entropy weight—fuzzy comprehensive evaluation method to confirm the weight of indices. The prominent feature of the entropy weight method is its strong objectivity, which means it depends on the discreteness of the

data itself and will not change with the subjectivity of researchers. The fuzzy comprehensive evaluation method transforms the qualitative evaluation into a quantitative evaluation according to the membership theory of fuzzy mathematics and constructs the fuzzy comprehensive evaluation matrix through expert scoring. The entropy weight—fuzzy comprehensive evaluation method can not only reflect the information contained in objective data but also fully respect the opinions of experts.

(1) Establishing the original relationship matrix.

The original relationship matrix $X = \{x_{ij}\}_{m \times n}$ is the evaluation objects $m$ corresponding to evaluation indices $n$, where $x_{ij}$ is the index value of project corresponding $j$ to the $i$th evaluation object.

(2) Data standardization processing. Each index has different dimensions, so it is necessary to process the index data to obtain the standardization matrix $Y = \{y_{ij}\}_{m \times n}$, as shown in Equation (1).

$$y_{ij} = \frac{x_{ij} - x_{jmin}}{x_{jmin} - x_{jmax}} \tag{1}$$

where $y_{ij}$ is the standard value.

(3) Calculate the membership moment, as shown in Equation (2).

$$p_{ij} = \frac{y_{ij}}{\sum_{i=1}^{m} y_{ij}} \tag{2}$$

where $p_{ij}$ is the characteristic proportion of the ith evaluation object according to index $j$.

(4) Calculate the information entropy and difference coefficient, as shown in Equation (3).

$$e_j = -\frac{1}{lnm} \sum_{i=1}^{m} ln p_{ij} \quad d_j = 1 - e_j \tag{3}$$

where $e_j$ is the information entropy of the $j$th index and $d_j$ is the difference coefficient of the $j$th index.

(5) Calculate the index weight, as shown in Equation (4).

$$w_j = \frac{d_j}{\sum_{i=1}^{m} d_j} \tag{4}$$

where $w_j$ is the weight of the $j$th index.

(6) Construct the fuzzy comprehensive evaluation matrix.

First, the evaluation level of decision-making comments is graded according to the needs, represented by $V = \{v_1, v_2, v_3 \cdots, v_k\}$. Based on expert scoring, the number of experts at each evaluation level divided by the total number of experts is used as the membership value of evaluation factors for assessment, and the membership matrix $R$ is constructed.

(7) Establish a comprehensive evaluation model.

$$B = W \times R \quad D = V \times B^{-1} \tag{5}$$

where $W$ is the weight set of 15 index factors calculated by the entropy weight method and fuzzy comprehensive evaluation matrix, $B$ is the comprehensive evaluation, $V$ is the comment set, and $D$ is the TOD comprehensive development index of the rail transit stations.

## 2.4. Site Type Division Based on Spatial Clustering Features

After analyzing the TOD measurement of different rail transit stations through the three dimensions of "Node," "Place" and "Connection," to better characterize the commonality of parking supply characteristics of similar stations, cluster analysis is needed. Cluster analysis generally extracts Z data features to obtain a map from basic data to Z-dimensional vector space to maximize the feature similarity of similar data. Among different clustering

methods, the K-means algorithm, as a dynamic clustering algorithm, has the characteristics of scalability, fast computing speed, and excellent clustering effect.

The principle of the K-means algorithm is to divide each data point into the cluster where the nearest clustering center is located, and the specific process of the cluster analysis of the TOD measurement is as follows:

(1) According to the TOD comprehensive development index set of rail transit stations obtained from the previous analysis, k data points are randomly selected as the initial clustering center point $D_i = (f_i, g_i, h_i)$.

In which $f_i, g_i, h_i$ represents the indices of the Node dimension, Place dimension, and Connection dimension.

(2) The Euclidean formula is used to calculate the distance $d(x_i, D_i)$ between the index of each site and the cluster center point and divide it into the nearest class, as shown in Equation (6):

$$d(x_i, D_i) = \sqrt{(x_i - f_i)^2 + (x_i - g_i)^2 + (x_i - h_i)^2} \tag{6}$$

(3) Recalculate the average value of data points in each category after change and take the obtained average value points as new cluster centers.

(4) Iterating steps (2)–(3) and stopping when each cluster no longer changes or reaches the maximum number of iterations.

## 3. Reduction Model of the Parking Allocation Index for Office Buildings Based on Different Clustering Methods

Generally, the calculation methods of parking allocations for office buildings are divided into two categories: the method based on survey and statistical analysis and the method based on parking demand prediction. The parking demand prediction method includes the prediction model based on land use, the prediction model based on vehicle travel relationship attraction, the prediction model based on correlation analysis [13]. Leung [14] determined the impact of the lower limit of the parking allocation index on land use based on the land area's population and vehicle statistics of off-road parking lots in Auckland. Paul [15] studied the quantification, revision, elasticity index, and index base of the parking allocation index of commercial and office buildings. Al-Sahili [16] established a parking generation model based on a parking survey of different types of buildings to formulate the parking requirements indices of auxiliary buildings. According to the parking generation rate model, Janak [17] proposed the problems related to and due to the parking, various parking characteristics and their applications, parking choice behavior of drivers, development of demand models considering various factors. Li [18] proposed a method to correct the parking demand calculated according to the existing allocation indices through the division of parking functions and the analysis of location condition attributes and established the corresponding correction model. Based on the prediction of the parking demand rate, Ma [19] proposed a method to formulate parking allocation index standards in residential areas based on the correction coefficient and combined it with the characteristics of the city. Davis, AY [20] studied the environmental and economic costs of sprawling parking lots in the United States from the perspective of parking classification, parking investigation, demand analysis, and parking allocation index strategy. Aiming at the problem of parking requirements and allocation of urban office buildings, Gong Y C [21] introduced the parking generation rate correction model of parking turnover rate and motor vehicle growth rate in peak hours and proposed the office building parking allocation model based on the parking demand and supply balance of traffic location analysis.

Most of the existing studies only consider the determination methods of the parking allocation index for office buildings under normal conditions. For office buildings under different development conditions, especially for TOD development-oriented office buildings, there is a lack of research on parking indices.

### 3.1. Basic Ideas of the Model

The parking allocation index of office buildings is mainly determined by parking allocation, and the parking allocation index and parking demand are restricted and influence each other. For an office building, the trip mode is relatively fixed, and the parking demand mainly depends on the proportion of car travel for daily commuting. In other conditions, in the office buildings in the influence area of rail transit stations with the TOD measure, under the public transportation travel orientation, the proportion of travelers choosing to travel by car will change, and under different measure levels, the attractiveness of public transportation to travelers and the variation range of the proportion of car travel are both different. The change in car travel proportion, which is the actual performance of the change in parking demand, is also the target that the reduction ratio needs to characterize.

The disaggregate model is widely used in the research on trip mode split. Its selection model selects the scheme with the greatest utility from multiple alternatives based on different influencing factors. Therefore, this paper describes the changes in travelers' trip modes based on the disaggregate model to build a reduction model of the parking allocation index for office buildings in the affected area of rail transit stations based on TOD measurements.

### 3.2. Method: Trip Mode Split Model

The disaggregated model is based on utility theory, and travelers always tend to choose the option with the greatest utility. Its utility function consists of two parts: the deterministic component $V_{ab}$ and the random component $\varepsilon_{ab}$, assuming that the random component and the deterministic component are independent of each other and obey the Gumbel function distribution. The deterministic function $V_{ab}$ usually adopts a linear function as its expression.

$$U_{ab} = V_{ab} + \varepsilon_{ab} \tag{7}$$

$$V_{ab} = \sum_{z=1}^{z} \theta_{az} X_{abz} = \theta_1 X_{ab1} + \theta_2 X_{ab2} + \cdots + \theta_z X_{abz} \tag{8}$$

In Equation (8), $X_{abz}$ is the $z$th characteristic variable contained in the $a$th option chosen by traveler $b$. $z$ is the number of characteristic variables. $\theta_z$ is the parameter corresponding to the $z$th characteristic variable.

Travel cost is not only an important factor in the trip mode split but also a measure of utility. This paper focuses on the impact of changes in the two modes of a trip, car and rail transit. Travelers from office buildings mainly travel for daily commuting, and the main factors affecting their car travel cost include travel time, fuel cost, and parking expenses (parking fees for parking lots built for office buildings). The main factors affecting their rail transit travel cost include walking distance (walking distance from rail station to office building), ticket price, and congestion cost. In addition, the personal characteristics of travelers will also affect the trip mode split. The factors affecting the time value of parking choice behavior was analyzed by Hu [22]; it was found that gender, age, the number of years of having a driving license, and income were the main factors of travelers. Sbh A [23] analyzed the choice of parking mode from different perspectives based on the MNL model, and the results show that the personal factors of gender, age, and monthly income had a significant impact on the choice of parking behavior. According to the relevant research analysis, the paper focuses on three possible variable factors: gender, age, and monthly income.

The multinomial logit model (MNL) is one of the most widely used models in the aggregate model and can describe the selection probability of multiple schemes. This paper uses the MNL model to construct a trip mode behavior selection model. For travelers from office buildings, their travel purpose is fixed. When external factors change, the trip mode of travelers can be summarized into three types: choosing cars in parking lots in office

buildings, cars in other parking lots, or rail transit. The probability of a traveler's trip mode split is shown in Equation (9):

$$P_{ab} = \frac{\exp(V_{ab})}{\sum_{c \in A_b} \exp(V_{cb})} \tag{9}$$

### 3.3. Parking Allocation Index Reduction Model for Office Building

The reduction proportion of parking allocations for office buildings is not only related to the TOD measurement level of rail transit stations but also depends on the walking distance. The reduction proportion corresponding to different categories is different, while the reduction proportion is also different for the same category and different spatial distances [24].

$$\nabla P_k = \frac{P_k' - P_k^0}{P_k^0} \tag{10}$$

In the formula:

$\nabla P_k$ is the reduction ratio of the parking allocation index of office buildings at a certain distance from k-type rail transit stations.

$P_k'$ is the proportion of car travel selected to build parking lots after the change in walking distance of k-type rail transit stations.

$P_k^0$ is the proportion of car trips at k-type rail transit stations currently selected to build parking lots.

## 4. Example Analysis

### 4.1. Description of the Research Area and Data

Nanjing is the capital city of Jiangsu Province, in which the first subway line was opened in 2005. By the end of 2021, the total length of the Nanjing subway was 427.1 km, with an annual passenger volume of 1.278 billion. The data used in this paper include four aspects. (1) The heat map is based on the mobile phone signaling data, which is used to reflect the movement and aggregation of population. (2) Spatial data is obtained from the Baidu map and Walk Score website and API data is based on Web crawler. (3) Nanjing City rail transit stations and relevant data are derived from the official website of Nanjing authorities. (4) Parking and trip data of office buildings around Nanjing rail transit stations are derived from a parking survey based on the engineering project "Standards and Guidelines for the Setting of Parking Facilities for Buildings in Nanjing", which is funded by the local government department.

### 4.2. Cluster Analysis of Rail Transit Stations Based on TOD Measurement

Before calculating the TOD measurement, it is necessary to define the influence area of the rail transit station. According to the guidelines for transit-oriented land-use development in Nanjing, the influence area of rail transit stations is defined according to a 10-min walking distance (approximately 800–1500 m radius). Yin Yue [25] proposed that the influence area of rail transit stations in Nanjing should be within a 1 km radius. Therefore, this paper takes the area within a 1 km radius of the rail station as the influence area of the rail transit station. The TOD measurement of Nanjing rail transit stations is measured from the three dimensions of Node, Place, and Connection, and the comprehensive score is calculated to obtain the measurement index of each station, as shown in Figure 1.

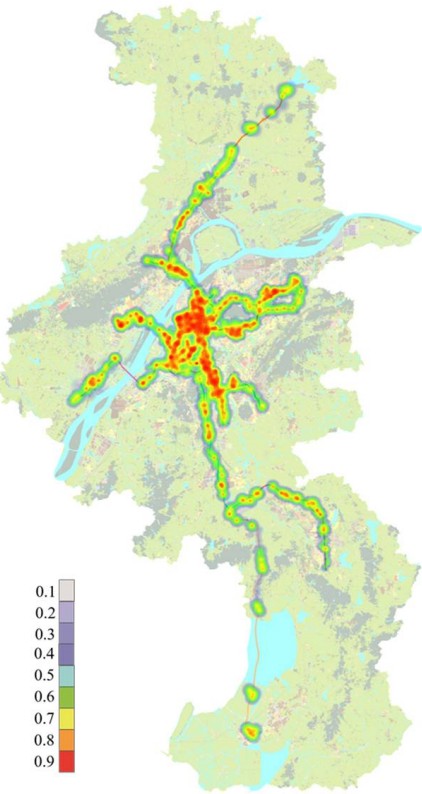

**Figure 1.** TOD Measurement Analysis Chart.

Generally, the comprehensive measurement level of stations in the old urban area (such as Xinjiekou Station) is high, which is served by multiple rail transit lines and has an outstanding service capacity and low information entropy. The comprehensive measurement level of stations in peripheral areas (such as Jiangning District and Pukou District) shows a decreasing trend, mainly because the service capacity of peripheral stations is weak, and the land type is relatively singular. For example, the land around Tianruncheng Station is basically residential land. However, there are stations with the second highest comprehensive measurements in the Xianlin area and Baijiahu area, indicating that rail transit strongly promotes local economic development.

According to the TOD measurement, cluster analysis is carried out on the stations, which is shown in Figure 2.

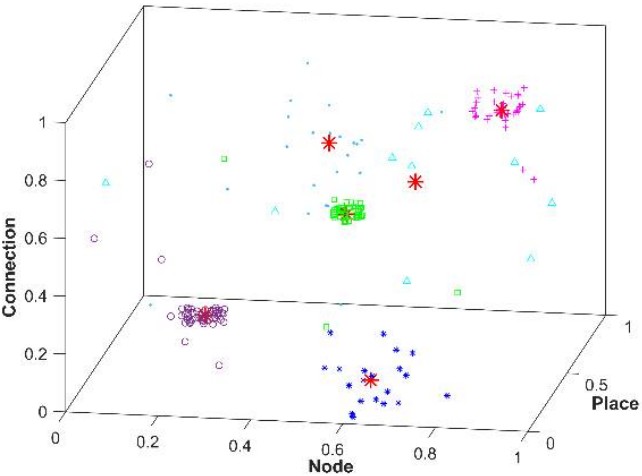

**Figure 2.** Cluster analysis of rail transit stations.

Based on the TOD measurement of each station, Nanjing rail transit stations are divided into four categories: Coordinated Stations, Prioritized Stations, Potential Stations and Imbalanced Stations, as shown in Figure 3. The Coordinated Stations are mainly located in the core area, with outstanding service capacity, rich land use, and high contact level. The Prioritized Stations are mainly distributed on each line radiating outward from the core area. The Potential Stations are scattered in each area, which has a low TOD measurement and much room for development. The Imbalance Stations are mainly distributed in the city periphery. Considering that the service capacity of the Imbalance Station and the Connection level with surrounding places is low, this paper does not consider the parking allocation index of office buildings around such stations.

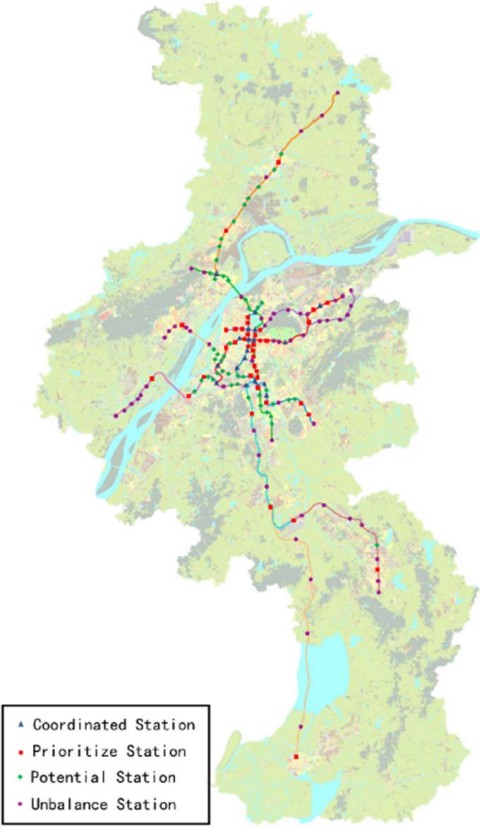

**Figure 3.** Classification map of rail transit stations.

### 4.3. Model of Trip Mode Split

In this paper, three typical stations, Xinjiekou Station, Huashenmiao Station, and Xinghuo Road Station, are selected for the Coordinated Station, Prioritized Station, and Potential Station. A trip mode split survey is carried out for the office buildings around each typical station, and more than 250 questionnaires are distributed at each station. This paper adopts the comprehensive survey method of the combination of a revealed preference (RP) survey and a stated preference (SP) survey, and the research includes three parts: (1) socioeconomic attributes: including residents' gender, age, and monthly income; (2) private car travel attributes: including travel time, parking cost, and fuel cost; and (3) trip split: investigate the parking choice intention of residents when the walking distance increases by 100 m (continue to choose the equipped parking lot, other parking lots or public transportation).

(1) Coordinated Station—Xinjiekou Station.

The main feature of such stations is that they have a high level of TOD measurement, are located in the core area of the city, and have a high level of transit development, population vitality, and land collaborative utilization. A total of 251 questionnaires were

distributed at Xinjiekou Station, and 208 valid questionnaires were recovered, with an effective recovery rate of 82.9%. After quantifying the questionnaire data, the MNL model was used to calibrate the parameters of the trip split model. The calibration results are shown in Table 2.

**Table 2.** Parameter Calibration Results.

| | Multinomial Logistic Regression | | | | Number of Obs = 208 LR Chi2(18) = 61.09 Prob > Chi2 = 0.0000 | |
|---|---|---|---|---|---|---|
| | | | Log Likelihood = −72.135961 | | Pseudo R2 = 0.6975 | |
| y | Coef. | Std. Err. | z | P > \|z\| | [95% Conf. | Interval] |
| 1 | (Base Outcome) | | | | | |
| 2 | | | | | | |
| a | −0.7803528 | 0.3623199 | −2.15 | 0.031 | −1.490487 | −0.0702188 |
| b | 1.47873 | 1.009419 | 1.46 | 0.143 | −0.4996957 | 3.457156 |
| c | 0.0088726 | 0.4288599 | 0.02 | 0.983 | −0.8316774 | 0.8494226 |
| d | 2.328356 | 0.6184704 | 3.76 | 0.000 | 1.116177 | 3.540536 |
| e | −0.2492888 | 0.2607421 | −0.96 | 0.339 | −0.7603339 | 0.2617562 |
| f | 1.040864 | 0.3737741 | 2.78 | 0.005 | 0.3082799 | 1.773448 |
| g | −0.9921391 | 0.4201079 | −2.36 | 0.018 | −1.815536 | −0.1687427 |
| h | −0.5698772 | 0.3764375 | −1.51 | 0.130 | −1.307681 | 0.1679267 |
| i | −0.190442 | 0.3222121 | −0.59 | 0.554 | −0.821966 | 0.4410821 |
| _cons | −0.5278871 | 3.576199 | −0.15 | 0.883 | −7.537108 | 6.481334 |
| 3 | | | | | | |
| a | −0.9065779 | 0.3368911 | −2.69 | 0.007 | −1.566872 | −0.2462834 |
| b | 1.625281 | 0.9499861 | 1.71 | 0.087 | −0.2366571 | 3.48722 |
| c | −0.1603834 | 0.4009301 | −0.40 | 0.689 | −0.9461919 | 0.6254251 |
| d | 1.966425 | 0.6033083 | 3.26 | 0.001 | 0.7839626 | 3.148888 |
| e | −0.2234546 | 0.2412923 | −0.93 | 0.354 | −0.6963788 | 0.2494695 |
| f | 0.8111466 | 0.3524506 | 2.30 | 0.021 | 0.1203561 | 1.501937 |
| g | −1.190372 | 0.4028082 | −2.96 | 0.003 | −1.979862 | −0.4008829 |
| h | −0.6734303 | 0.3573285 | −1.88 | 0.059 | −1.373781 | 0.0269207 |
| i | −0.1822351 | 0.3057513 | −0.60 | 0.551 | −0.7814966 | 0.4170263 |
| _cons | 3.254876 | 3.121613 | 1.04 | 0.297 | −2.863372 | 9.373124 |

In the calibration results, the log likelihood of the model is −72.14, the likelihood ratio test statistic (LR chi2) is 61.09, and the corresponding *p* value is 0, indicating that the model is significant. The goodness of fit (Pseudo R2) of the model is 0.6975, indicating that the fitting effect of the model is good. Therefore, the probability of trip mode split in office buildings around such stations is shown in Equation (11):

$$
\begin{aligned}
ln(\tfrac{P_2}{P_1}) &= -0.528 - 0.78X_1 + 1.479X_2 + 0.009X_3 + 2.328X_4 - 0.249X_5 \\
&\quad + 1.041X_6 - 0.992X_7 - 0.57X_8 - 0.19X_9 \\
ln(\tfrac{P_3}{P_1}) &= 3.255 - 0.907X_1 + 1.625X_2 - 0.16X_3 + 1.966X_4 - 0.223X_5 \\
&\quad + 0.811X_6 - 1.19X_7 - 0.673X_8 - 0.182X_9
\end{aligned}
\tag{11}
$$

(2) Prioritized Station—Huashenmiao Station.

The main feature of such stations is that the TOD measurement level is moderate since the stations are generally located outside the city's core area. The levels of transit development, population vitality, and land collaborative utilization are moderate, and there is still some room for development. A total of 250 questionnaires were distributed at Huashenmiao Station, and a total of 201 valid questionnaires were recovered, with an effective recovery rate of 80.4%. After quantifying the questionnaire data, the parameters

of the trip mode split model were calibrated. For the office buildings around such stations, the car trip mode split probability is shown in Equation (12):

$$
\begin{aligned}
ln(\tfrac{P_2}{P_1}) = {} & -8.015 + 3.097X_1 - 1.97X_2 - 4.406X_3 + 3.549X_4 \\
& + 1.917X_5 + 0.326X_6 - 0.983X_7 - 1.527X_8 \\
& - 2.253X_9 \\
ln(\tfrac{P_3}{P_1}) = {} & -12.173 + 3.651X_1 - 2.646X_2 - 5.501X_3 + 4.469X_4 \\
& + 2.7X_5 + 0.406X_6 - 0.656X_7 - 1.779X_8 - 1.584X_9
\end{aligned}
\tag{12}
$$

(3) Potential Station—Xinghuo Road Station.

The main feature of such stations is that the TOD measurement level is low, and the transit development level, population vitality, and land collaborative utilization are low. A total of 250 questionnaires were distributed at Xinghuo Road Station, and 199 valid questionnaires were recovered, with an effective recovery rate of 79.6%. After quantitative processing of the questionnaire data, the parameters of the trip mode split model were calibrated. The trip probability of cars in the office buildings around such stations is shown in Equation (13):

$$
\begin{aligned}
ln(\tfrac{P_2}{P_1}) = {} & 19.252 - 1.256X_1 + 1.509X_2 - 1.171X_3 + 2.267X_4 - 1.242X_5 \\
& + 0.521X_6 - 3.109X_7 - 0.672X_8 - 0.704X_9 \\
ln(\tfrac{P_3}{P_1}) = {} & 34.639 - 3.875X_1 + 3.543X_2 - 2.095X_3 + 5.584X_4 - 2.885X_5 \\
& + 1.348X_6 - 8.09X_7 - 1.756X_8 - 1.861X_9
\end{aligned}
\tag{13}
$$

### 4.4. Analysis of Parking Allocation Index Reduction for Office Buildings

According to the models built for the above three types of stations, Figure 4 shows the variation curve of the car share rate under different walking distances.

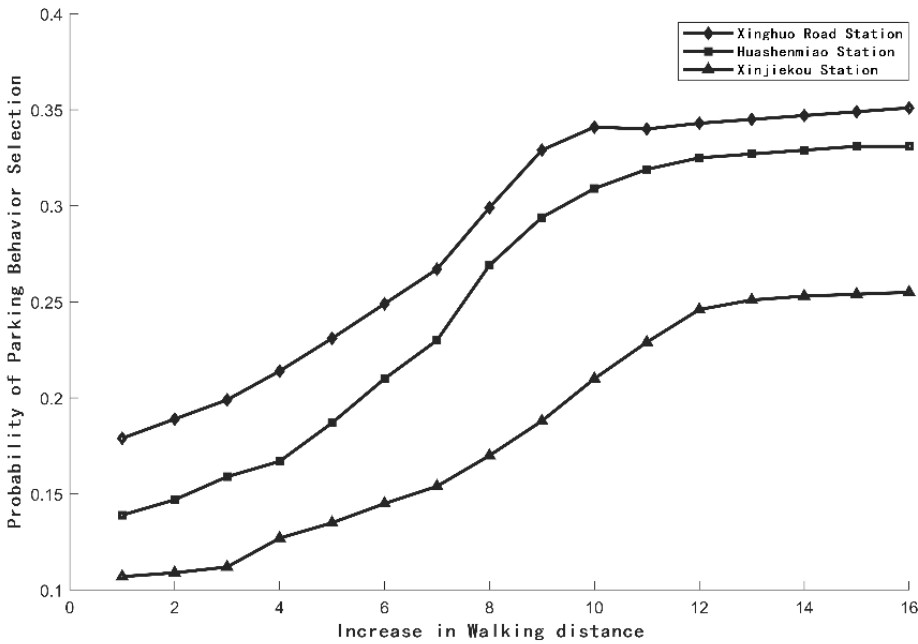

**Figure 4.** Variation curve of the car share rate under different walking distances.

Using the constructed parking allocation index reduction model for office buildings, the reduction ratio of the parking allocation index for office buildings in the influence areas of different types of stations under different walking distances is calculated, as shown in Table 3.

**Table 3.** Reduction ratio of the parking allocation index for office buildings.

| Station Name / Walking Distance | Xinjiekou Station | Huashenmiao Station | Xinghuo Road Station |
|---|---|---|---|
| <100 m | 42% | 30% | 23% |
| 100–200 m | 40% | 28% | 23% |
| 200–300 m | 39% | 27% | 21% |
| 300–400 m | 35% | 20% | 18% |
| 400–500 m | 32% | 19% | 14% |
| 500–600 m | 30% | 18% | 11% |
| 600–700 m | 24% | 16% | 9% |
| 700–800 m | 23% | 11% | 8% |
| 800–900 m | 18% | 10% | 6% |
| 900–1000 m | 12% | 5% | 1% |

## 5. Discussion

Taking Nanjing as an example, based on the cluster analysis of TOD measurement, this paper selects three different types of rail transit stations, and gives the reduction proportion of parking index of office buildings at different distances from rail transit stations. According to the previous analysis results, it can be seen that:

(1) The reduction coefficient of the parking allocation index of office buildings around the three types of stations gradually decreases with increasing walking distance, and the degree of reduction increases. Under the same walking distance, the higher the TOD measure of rail stations is, the greater the reduction proportion, that is, coordinated development type > key incubation type > potential development type.

(2) For Coordinated Stations, the reduction in the parking allocation index of office buildings with a walking distance of less than 400 m is less sensitive to distance, and its value changes little. Within the range of 400–700 m, the parking allocation index is reduced by approximately 4%. However, after more than 700 m, the sensitivity to distance becomes stronger, and the reduction in parking configuration decreases by approximately 7% for every 100 m.

(3) For Prioritized Stations, the reduction in the parking allocation index of office buildings with a walking distance of less than 400 m is less sensitive to distance, and the reduction in the parking allocation index shows little change. In the range of 400–700 m, the parking allocation index gradually decreases, and its sensitivity gradually becomes stronger. When it is more than 700 m, the change is more obvious.

(4) For Potential Stations, the sensitivity of parking allocation index reduction to distance is always strong. In the process of gradually increasing walking distance, the sensitivity gradually becomes stronger, and the change is more obvious when the walking distance is more than 500 m.

Therefore, the calculation of parking allocations of office buildings around rail stations based on different TOD measurements can not only provide quantitative index calculation methods at different distances but also effectively guide residents to choose green trip modes and are useful to the construction of sustainable transportation systems.

## 6. Conclusions

The reduction of parking allocations around TOD stations will effectively guide green trip modes and enhance the sustainable development of transportation in the surrounding areas. This paper, based on the classical "Node-Place" model, adds the "Connection" dimension, puts forward the TOD measurement method of rail transit stations and analyzes it by the k-means cluster algorithm. According to the clustering results, the trip mode split model is constructed based on the MNL model, and the reduction calculation method of the parking allocation index of office buildings in the influence area around rail transit stations is proposed. According to the Nanjing case study, the reduction model of the parking allocation index constructed in this paper can identify the measurement of the

rail transit stations well and the reduction proportion of the parking allocation index under different types of office buildings and various walking distances. This is given to promote the proportion of green trip modes in surrounding areas and ensure the sustainable development of transportation in core areas.

**Author Contributions:** Conceptualization, J.M. and X.T.; methodology, X.T.; software, P.H.; validation, C.X., P.H. and X.T.; formal analysis, J.M.; investigation, J.M. and X.T.; resources, P.H.; data curation, C.X.; writing—original draft preparation, C.X.; writing—review and editing, P.H.; visualization, P.H.; supervision, J.M.; project administration, J.M.; funding acquisition, J.M. All authors have read and agreed to the published version of the manuscript.

**Funding:** This research received no external funding.

**Institutional Review Board Statement:** Not applicable.

**Informed Consent Statement:** Not applicable.

**Data Availability Statement:** Not applicable.

**Conflicts of Interest:** The authors declare no conflict of interest.

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
