# Peer review of "Parking Allocation Index Analysis of Office Building Based on the TOD Measurement Method"

_sustainability, doi:10.3390/su14052482_

Round 1
Reviewer 1 Report
This paper developed a new method by the TOD Measurement based on Sustainable Transportation Development to Parking Allocation of Office Building.
The application results showed the superiority of the model and an illustrative example was included to illustrate the computational procedure. Anyway, I have some comments and points that are required to be addressed as the follows.
1、In Line 322, 1 or 1)? In Line 336, 2 or 2)? In Line 345, 3 or 3)?
2、In Line 256, In Line 358, In Line 260, “=” is “is”.
3、I think some suggestions should be provided after analysis, which can display the significance of the method you proposed.
4、The language needs to be polished.
Author Response
Reviewer 1:
1、In Line 322, 1 or 1)? In Line 336, 2 or 2)? In Line 345, 3 or 3)?
REPLY: We have deleted the label 1, 2, 3.
2、In Line 256, In Line 358, In Line 260, “=” is “is”.
REPLY: We have changed “=” to “is”.
3、I think some suggestions should be provided after analysis, which can display the significance of the method you proposed.
REPLY: We have added a “Discussion” section to give more comprehensive description of the research results of our paper.
4、The language needs to be polished.
REPLY: This article has been polished by a professional polishing organization. The Editing Certificate of the professional organization was in the attachment.

Reviewer 2 Report
Interesting paper. In the reviewed paper, the Authors selected office buildings in the rail transit station influence area as the research object, puts forward the transport oriented development measurement method of rail transit stations based on the improved "Node-Place" model, and clusters the stations under different measurement indices by the K-means algorithm. For different types of stations, the multinomial logit model was used to build different types of trip mode split models to put 20 forward the reduction calculation method of the parking allocation index of office buildings in the rail transit station influence area. Finally, in this paper, the Authors taken the revision of Nanjing's allocation index in 2019 as an example, and the transport-oriented development measurement was identified through the "Node-Place-Connection" model. The Authors proposed the optimized calculation method of the parking allocation index for office buildings. The results indicated that the method can reduce parking allocations to encourage the use of green transportation and guide the construction of urban sustainable transportation systems. In my opinion, the paper can be published, after taking into account the following remarks:
- line 31: what does it mean "IEA statistics"? All acronyms used in the paper for the first time should be at the same time explained,
- at the end of the Introduction section, the Authors should shortly write what was the main aim of this paper, as well as what was contained in each paper section,
- The main aim of this paper is the parking allocation index calculation method of office buildings by the transport-oriented development measurement based on sustainable transportation development. Authors in the Introduction section discuss the key findings which inform among others about the emissions of pollutants generated by the transport sector, the demand for parking lots for buildings built near railway stations, the allocation indicators for parking lots for office buildings, etc. This is very good, but unfortunately, the Authors did not mention the other existing infrastructure solutions that support the efficient functioning of transport oriented development and sustainable transportation like park and ride solutions, which anable to leave the private car at the train station and onward journey (e.g. to the city center) with the use of means of public transport, e.g. railway. Authors should mention these worldwide popular solutions and refer to the latest scientific literature on the subject in this regard, i.e. "The analysis of the factors determining the choice of park and ride facility using a multinomial logit model", 10.3390 / en14010203; "A P-Hub Location Problem for Determining Park-and-Ride Facility Locations with the Weibit-Based Choice Model", doi.org/10.3390/su13147928; "P&R parking and bike-sharing system as solutions supporting transport accessibility of the city", doi 10.21307 / TP-2020-066. One short paragraph in the Introduction section will be enough,
- lines from 126 to 136 (and other similar cases in the whole paper text): The Authors used the numbers 1,2,3,4. Usually, in the scientific paper, the numbers are reserved for indicating the sections and subsections, so, in this case, it is recommended to replace the numbers with bullet items,
- equations from (1) to (4), there is a lack of explanation of some used variables, e.g. dj. The Authors should check in the whole paper and add the explanation of all used acronyms/variables in the equations,
- there is a lack of the "Materials and Method" section, which is recommended by Sustainability journal paper requirements,
- lines from 239-240, the Authors wrote like follows: ..."This paper focuses on three possible variable factors: gender, age, and monthly income." How these variables were selected to further analysis?
- lines 271-272, the Authors wrote like follows "Parking and trip data of office buildings around Nanjing rail transit stations are derived from parking survey. ", but there is, unfortunately, a lack of any detailed information about this research survey,
- Figure 2. Cluster analysis of rail transit stations: lack of names and units of axis "x" and axis "y". It should be improved,
- there is a lack of "Discussion" section. In this section, the Authors should present a detailed discussion dedicated to obtained results,
- is "5 Conclusion" should be "5 Conclusions".
Author Response
1、Line 31: what does it mean "IEA statistics"? All acronyms used in the paper for the first time should be at the same time explained.
REPLY: We have added an explanation of "IEA" in Line31.
2、At the end of the Introduction section, the Authors should shortly write what was the main aim of this paper, as well as what was contained in each paper section.
REPLY: At the end of the introduction, we add an introduction to the structure of the paper and what was contained in each paper section in lines 65-69.
3、The Authors did not mention the other existing infrastructure solutions that support the efficient functioning of transport oriented development and sustainable transportation like park and ride solutions, which anable to leave the private car at the train station and onward journey (e.g. to the city center) with the use of means of public transport, e.g. railway. Authors should mention these worldwide popular solutions and refer to the latest scientific literature on the subject in this regard, i.e. "The analysis of the factors determining the choice of park and ride facility using a multinomial logit model", 10.3390 / en14010203; "A P-Hub Location Problem for Determining Park-and-Ride Facility Locations with the Weibit-Based Choice Model", doi.org/10.3390/su13147928; "P&R parking and bike-sharing system as solutions supporting transport accessibility of the city", doi 10.21307 / TP-2020-066. One short paragraph in the Introduction section will be enough.
REPLY: In the introduction part of the paper, the literature review of the current sustainable transportation solutions at home and abroad is supplemented, including the relevant literature mentioned by the reviewer, such as "A P-Hub Location Problem for Determining Park-and-Ride Facility Locations with the Weibit-Based Choice Model","P& R parking and bike-sharing system as solutions supporting transport accessibility of the city".
4、Lines from 126 to 136 (and other similar cases in the whole paper text): The Authors used the numbers 1,2,3,4. Usually, in the scientific paper, the numbers are reserved for indicating the sections and subsections, so, in this case, it is recommended to replace the numbers with bullet items.
REPLY: We have replaced the numbers 1,2,3,4 with 2-1, 2-2, 2-3 and 2-4, other similar cases also have been changed.
5、Equations from (1) to (4), there is a lack of explanation of some used variables, e.g. dj. The Authors should check in the whole paper and add the explanation of all used acronyms/variables in the equations.
REPLY: We have checked and added explanations to all the variables involved in the formula in the whole paper.
6、There is a lack of the "Materials and Method" section, which is recommended by Sustainability journal paper requirements.
REPLY: The method of this paper is mainly by constructing the reduction calculation model of parking index, the paper does not set a separate "Method" section, but the content is provided in Section 2 and Section 3. Section 2 introduces the TOD measurement method, which mainly constructs an improved" node-place "model and uses K-means algorithm for cluster analysis. Section 3 constructs a trip mode selection model, which mainly uses MNL model. In order to clarify the research method of the paper, we have added a short description in the Introduction.
7、Lines from 239-240, the Authors wrote like follows: ..."This paper focuses on three possible variable factors: gender, age, and monthly income." How these variables were selected to further analysis?
REPLY: According to the existing relevant research, in the construction of trip mode behavior selection model by running MNL, gender, age and monthly income are three variables representing personal characteristics, which are the main factors affecting travel behavior. We have added the explanation of variable selection in Lines from 257-264.
8、Lines 271-272, the Authors wrote like follows "Parking and trip data of office buildings around Nanjing rail transit stations are derived from parking survey. ", but there is, unfortunately, a lack of any detailed information about this research survey,
REPLY: The parking and travel data of office buildings around Nanjing rail transit stations are mainly based on the engineering project "Standards and Guidelines for the Setting of Parking Facilities for Buildings in Nanjing", which is funded by the local government department. And the author is in charge of the project. During the project, the parking and travel data of office buildings around rail transit stations were investigated, and a research report was formed.
9、Figure 2. Cluster analysis of rail transit stations: lack of names and units of axis "x" and axis "y". It should be improved.
REPLY: The Figure 2 is modified and added the names and units of axis "X" and axis "Y".
10、There is a lack of "Discussion" section. In this section, the Authors should present a detailed discussion dedicated to obtained results.
REPLY: Section 5 "Discussion" is added in the paper, the discussion dedicated to obtained results is provided in the section.
11、"5 Conclusion" should be "5 Conclusions".
REPLY: We have revised the name according to the opinions.
